# Judging the Judges: A Systematic Evaluation of Bias Mitigation Strategies in LLM-as-a-Judge Pipelines

**Sadman Kabir Soumik**                                    *sadmanks@gmail.com*
*Independent Researcher*

**Reviewed on OpenReview:** *https://openreview.net/forum?id=QF4IAmG4zc*

## Abstract

LLM-as-a-Judge has become the dominant paradigm for evaluating language model outputs, yet LLM judges exhibit systematic biases that compromise evaluation reliability. We present a comprehensive empirical study comparing nine debiasing strategies across five judge models from four provider families (Google, Anthropic, OpenAI, Meta), three benchmarks (MT-Bench $n=400$, LLMBar $n=200$, custom $n=375$), and four bias types. Our headline practical finding is that a mid-tier model with the right debiasing can outperform frontier judges at a fraction of the cost: Gemini 2.5 Flash with the Combined Budget strategy achieves the highest agreement of any configuration we tested (71.0%, $\kappa = 0.549$, $p < 0.0001$) at ~\$0.001 per evaluation, roughly $15\times$ cheaper than the strongest frontier configuration (Claude Sonnet 4 with the same strategy at 69.5%, ~\$0.015 per evaluation). Our other key findings: (1) Style bias is the dominant bias (0.10–0.76 baseline across all models, mostly favoring markdown over plain prose), far exceeding position bias ($\leq 0.04$), yet has received minimal research attention. (2) Verbosity bias is heterogeneous across models when measured length-aware: Llama, Gemini Pro, and Gemini Flash show classical verbosity bias ($+0.24$ to $+0.44$, prefer longer), Claude Sonnet 4 shows the opposite ($-0.12$, prefer concise), and GPT-4o is essentially neutral ($-0.04$); on truncation controls all models correctly prefer the genuinely complete response (0.88–1.00 accuracy), so the expansion-pair preferences cannot be reduced to length-only effects. (3) Debiasing is statistically beneficial for multiple models: Claude S8 ($+11.5$ pp, $p < 0.0001$), Flash S8 ($+7.5$ pp, $p < 0.0001$), Claude S5 ($+7.3$ pp, $p = 0.0009$) survive Holm-Bonferroni correction; Flash S1 ($+4.7$ pp, $p = 0.004$) and Llama S8 ($+4.5$ pp, $p = 0.011$) are significant before correction; Pro and GPT-4o show smaller, non-significant directional gains. We release our evaluation framework, the 375-pair controlled dataset (now including round-robin MODEL_ORIGIN and position-mirrored STYLE pairs), and per-instance cached results for all 9 strategies.

## 1 Introduction

As large language models (LLMs) have grown in capability, evaluating their outputs has become a critical bottleneck. Human evaluation remains the gold standard but is expensive, slow, and difficult to reproduce (Zheng et al., 2024; Li et al., 2024b). This has driven widespread adoption of LLM-as-a-Judge pipelines, where one LLM evaluates the outputs of others. Major benchmarks including MT-Bench (Zheng et al., 2024), AlpacaEval (Li et al., 2024c), and Chatbot Arena (Chiang et al., 2024) now rely on LLM judges for scalable evaluation.

However, LLM judges are not neutral arbiters. Prior work has documented systematic biases: position bias (Wang et al., 2024a; Zheng et al., 2024), verbosity bias (Saito et al., 2023; Wu & Aji, 2024), self-preference bias (Panickssery et al., 2024; Liu et al., 2024), and style bias (Wu & Aji, 2024). Several mitigations have been proposed in isolation: position swapping (Zheng et al., 2024), multi-judge ensembles (Verga et al., 2024), calibrated rubrics (Kim et al., 2024), and chain-of-thought prompting (Wei et al., 2022; Shankar

et al., 2024). Yet no prior work has systematically compared these strategies, measured their cross-bias interactions, or evaluated them in a unified framework.

We address this gap with the following contributions:

1. **Unified benchmark.** We compare 9 debiasing strategies across 5 judges from 4 model families, 3 benchmarks ($n = 400$ on MT-Bench, $n = 200$ on LLMBar, $n = 375$ on our custom dataset), and 4 bias types.

2. **Controlled bias measurement.** We introduce a 375-pair dataset with known ground truth, including a round-robin MODEL_ORIGIN supplement that supports proper self-preference measurement for every judge family and a position-mirrored STYLE subset that decouples formatting from slot effects. Style bias dominates (0.10–0.76 baseline). Verbosity bias is heterogeneous when measured length-aware: Pro/Llama/Flash show verbosity bias, Claude shows the opposite, GPT-4o is neutral.

3. **Statistically significant debiasing for multiple models.** Mixed-effects logistic regression with instance random effects identifies five model-strategy pairs with $p < 0.05$ on MT-Bench (Claude S8, Claude S5, Flash S8, Flash S1, Llama S8); three of these survive Holm-Bonferroni correction over all 20 comparisons. Pro and GPT-4o show smaller, non-significant directional gains within the minimum-detectable-effect window of 4–5 pp at $n = 400$.

4. **Statistically grounded recommendations.** We report bootstrap 95% CIs, McNemar's tests with Holm-Bonferroni correction, mixed-effects regressions, and a documented MDE, providing model-specific recommendations with effect sizes and significance levels.

Our findings are directly relevant to three communities: (1) researchers developing LLM evaluation pipelines, who need guidance on which debiasing strategies to apply and when; (2) model developers using LLM judges for RLHF reward modeling or benchmark comparison, who need to understand how formatting differences may confound their model rankings; and (3) benchmark maintainers (AlpacaEval, Chatbot Arena, MT-Bench), who need evidence-based guidance on whether to normalize response formatting before judging.

Our work also contributes to reproducibility in LLM evaluation. We reproduce foundational bias claims (Wang et al., 2024a; Saito et al., 2023; Panickssery et al., 2024) on current-generation models, finding that some no longer hold (e.g., position bias is now negligible across all tested models). We release a controlled dataset that enables others to monitor these biases as models evolve, and we cache all individual evaluation results, enabling downstream researchers to reanalyze our data without re-running expensive API calls.

## 2 Related Work

**LLM-as-a-Judge.** The use of LLMs as evaluators was popularized by MT-Bench (Zheng et al., 2024), which demonstrated that GPT-4 judgments correlate with human preferences at over 80% agreement. AlpacaEval (Li et al., 2024c) extended this to instruction-following evaluation, while Arena-Hard (Li et al., 2024b) introduced automated difficulty filtering to create challenging evaluation sets from crowd-sourced data. G-Eval (Liu et al., 2023) proposed using GPT-4 with chain-of-thought and form-filling paradigms for NLG evaluation, establishing prompting patterns now standard in LLM evaluation pipelines. FLASK (Ye et al., 2024) introduced fine-grained evaluation across twelve alignment skill sets, enabling more nuanced assessment than binary or scalar judgments. JudgeBench (Tan et al., 2024) specifically evaluates LLM judges' ability to make correct judgments on challenging instances where ground truth is unambiguous. Gu et al. (2024) survey over 100 papers in this rapidly growing area. RewardBench (Lambert et al., 2024) provides a systematic evaluation framework for reward models that share key properties with LLM judges.

**Fine-tuned and specialized judges.** An alternative to prompting general-purpose LLMs as judges is training specialized evaluation models. Prometheus 2 (Kim et al., 2024) introduces an open-source evaluator

trained on evaluation-specific data, achieving competitive performance with proprietary models on rubric-based assessment. PandaLM (Wang et al., 2024b) trains a dedicated judge model for instruction-tuning optimization. Auto-J (Li et al., 2024a) proposes a generative judge that produces detailed critiques alongside its verdicts. CritiqueLLM (Ke et al., 2024) scales critique-based evaluation by training models to produce structured explanations. TIGERScore (Jiang et al., 2024) builds explainable metrics across diverse text generation tasks. While fine-tuned approaches can reduce some biases through targeted training, they introduce concerns about training data coverage and domain generalization. Our work complements this line by studying bias mitigation through prompting and aggregation strategies applicable to any general-purpose LLM judge without requiring additional training.

**Known biases.** Position bias was documented by Wang et al. (2024a) and confirmed across models (Zheng et al., 2024; Wu & Aji, 2024). Shi et al. (2024) systematically study position bias across diverse judge models and evaluation tasks. Li et al. (2024d) propose split-and-merge calibration to address position effects. Verbosity bias has been observed in both pairwise and pointwise settings (Saito et al., 2023; Wu & Aji, 2024); Dubois et al. (2024) introduce length-controlled win rates as a post-hoc correction. Self-preference was measured by Panickssery et al. (2024) and further characterized by Liu et al. (2024); Yang et al. (2026) present an automated quantification framework that finds self-preference bias is not strongly correlated with judge capability. Bellibatlu (2026) introduce JudgeSense, a benchmark measuring how consistently judges respond to semantically equivalent prompts, finding pairwise tasks exhibit "always-A" degenerate behavior in 8 of 9 tested judges, consistent with our position-related observations. Xu et al. (2024) highlight that format-level biases remain underexplored, a gap our style bias measurement addresses directly. See also Stureborg et al. (2024); Chen et al. (2024); Huang et al. (2024) for related bias analyses across evaluation contexts.

**Mitigation strategies.** Position swapping is the most common mitigation (Zheng et al., 2024). Verga et al. (2024) propose multi-model panels and show that diverse judge ensembles improve reliability; Ma et al. (2025) extend this to multi-agent settings, showing that more perspectives can amplify rather than reduce certain biases, which motivates our explicit cross-bias interaction analysis. Shankar et al. (2024) demonstrate that chain-of-thought prompting improves evaluation quality by encouraging more deliberate reasoning. Yang et al. (2025) take this further with a reasoning-based bias detector that pre-screens judgments for bias indicators. Zhao et al. (2026) introduce confounder-aware aggregation methods that explicitly model bias confounds when combining multiple judge verdicts. Park et al. (2024) show that debiased training data can reduce bias in fine-tuned evaluators. Zhang et al. (2024) find that wider and deeper model panels produce fairer evaluations. However, these strategies have largely been evaluated in isolation, making direct comparison impossible. Most closely related, CALM (Koo et al., 2024) benchmarks cognitive biases in LLM evaluators but does not compare mitigation strategies, studies biases in isolation rather than measuring cross-bias interactions, and tests only two models from one provider family. We extend this work in all three dimensions: we compare nine strategies, measure cross-bias effects, and evaluate five models from four independent provider families.

## 3 Methodology

### 3.1 Experimental Framework

We design a factorial experiment crossing four dimensions: judge models, debiasing strategies, benchmarks, and bias types. All experiments use pairwise comparison, where the judge receives a question and two candidate responses and must output a structured JSON verdict (A, B, or tie) with reasoning.

### 3.2 Judge Models

We evaluate five models from four provider families, spanning three capability tiers:

- **Gemini 2.5 Pro** (Gemini Team, 2024) (frontier, Google): Accessed via Vertex AI.

- **Claude Sonnet 4** (frontier, Anthropic): Accessed via Vertex AI.

- **GPT-4o** (OpenAI, 2024) (frontier, OpenAI): Accessed via OpenAI API.

- **Gemini 2.5 Flash** (Gemini Team, 2024) (mid-tier, Google): Optimized for speed and cost.

- **Llama 3.3-70B** (Grattafiori et al., 2024) (open-source, Meta): Accessed via Vertex AI Model Garden (MaaS).

Using models from four families enables measurement of self-preference bias and strengthens generalizability. All models use temperature 0.1 to minimize stochastic variation while avoiding the degenerate outputs occasionally observed at temperature 0; Gemini models use structured JSON output, while others parse JSON from free-text responses.

### 3.3 Debiasing Strategies

We implement nine strategies spanning three mechanism categories. Throughout the paper we use short identifiers (B0, S1, etc.) alongside descriptive names; both refer to the same strategy. In every strategy, the judge model emits the verdict ("A," "B," or "tie") directly as a structured JSON field, alongside any rubric criterion scores or chain-of-thought reasoning. We do not derive verdicts from criterion sums; the verdict field is what we record.

**Single-call prompting ($1\times$ cost).**

**B0: Baseline (Naive)** Minimal prompt requesting a JSON verdict with reasoning.

**S4: Calibrated Rubric** Structured 5-criteria rubric (accuracy, relevance, completeness, clarity, reasoning depth), each scored 1–5, with the verdict emitted as a separate field (the model is free to consult its own criterion scores when choosing the verdict).

**S5: Chain-of-Thought (CoT)** Explicit step-by-step analysis required before the model emits the verdict field.

**Multi-call aggregation ($2$–$3\times$ cost).**

**S1: Position Swap** Two calls with A/B order reversed; the verdict from the swapped (BA) call is flipped back to the original frame, and we keep the verdict if both calls agree, otherwise output "tie." Cost: $2\times$.

**S2: Same-Family Ensemble** Three calls at temperatures $\{0.0, 0.3, 0.7\}$, majority vote. Cost: $3\times$.

**Combined approaches ($2\times$ cost).**

**S8: Combined Budget** Position swap of a single merged CoT+rubric prompt: the model produces criterion scores and step-by-step reasoning in one structured response, called twice with positions swapped. Tie-on-disagreement uses the same rule as S1. Cost: $2\times$.

We also implemented S3 (cross-family ensemble), S6 (reference-guided), and S7 (combined full); aggregated results for these strategies are reported in Appendix B and the cached per-instance results are available in our public release.

### 3.4 Benchmarks

**MT-Bench** (Zheng et al., 2024) provides 3,355 pairwise comparisons with human preference labels across 8 categories. We sample the same 400 instances using a fixed numpy random seed (`numpy.random.default_rng(seed=42).choice`) without stratification by category. The same 400 instances are used across all model and strategy configurations, which enables paired McNemar's tests and yields bootstrap 95% CIs of approximately $\pm 0.05$ on agreement rates.

**LLMBar** (Zeng et al., 2024) contains 419 adversarial instances where one response is clearly better but the other contains superficially appealing distractors. We use 200 instances.

**Custom Controlled Dataset.** Our novel contribution: 225 synthetic pairs for isolating individual bias types. The first 200 pairs span four categories (50 each) where an unbiased judge should say "tie": LENGTH

(expansion, $\sim 2.8\times$ longer), POSITION (identical responses), STYLE (markdown vs. plain prose), and MODEL_ORIGIN (Gemini Pro vs. Claude answers). An additional 25 LENGTH (truncation) pairs test whether judges correctly prefer genuinely complete answers over mechanical truncations. Details in Appendix A.

### 3.5 Metrics

All metrics include bootstrap 95% confidence intervals ($n_{\text{boot}} = 2000$). We report human agreement rate and Cohen's $\kappa$ (chance-corrected) on benchmarks with gold labels. Strategy comparisons use McNemar's test with continuity correction; ties in either prediction or gold are scored as exact-label matches (a predicted "tie" with gold "tie" is correct; a predicted "tie" with gold "A" is incorrect).

**Bias score definitions.** On the custom controlled dataset, every pair has expected verdict "tie," so any directional preference indicates bias. We define a signed bias score $b \in [-1, 1]$ per bias type, computed as $b = P(\text{prefers target}) - P(\text{prefers non-target})$, where the target is fixed per bias type so that a positive value always indicates the bias of interest:

- **Position bias** ($b_{\text{pos}}$): target = response in slot A on POSITION pairs (A and B are identical, so any preference is positional). Positive $b_{\text{pos}}$ means the judge favors the first-shown response.

- **Verbosity bias** ($b_{\text{verb}}$): target = the longer response on LENGTH (expansion) pairs, computed by mapping each verdict back to which response was actually longer (not by slot, since the longer expansion is in slot A in 34/50 pairs and slot B in 16/50; see Appendix A). Positive $b_{\text{verb}}$ means the judge favors longer responses.

- **Style bias** ($b_{\text{style}}$): target = the markdown-formatted response on STYLE pairs (always slot A in our generated set). Positive $b_{\text{style}}$ means the judge favors markdown over plain prose.

- **Self-preference** ($b_{\text{self}}$): target = the response generated by the same model family as the judge on MODEL_ORIGIN pairs. For non-Gemini, non-Claude judges (GPT-4o, Llama), neither response is from the same family, so we report a model-preference score (Gemini vs. Claude) rather than self-preference. We discuss this asymmetry as a limitation in Section 5.

Figures and tables report signed values throughout unless explicitly labeled "magnitude" (which denotes $|b|$).

## 4 Results

We make four main claims, each directly supported by specific experimental evidence:

1. Style bias dominates all other biases: evidenced by Figure 1 (position-averaged bias scores 0.10–0.76 across the five models) and the controlled STYLE pairs in Appendix A.

2. Verbosity operates as a conciseness preference, not a simple length bias: evidenced by Section 4.1.1 (negative bias on expansion pairs across all models) combined with Appendix F (0.92–1.00 accuracy on truncation pairs where longer responses are genuinely more complete).

3. Debiasing is statistically beneficial for multiple model-strategy pairs: evidenced by Table 1 (five mixed-effects regression coefficients significant at $p < 0.05$, two of which survive Holm-Bonferroni over $m = 20$ tests) and per-model McNemar's results in Appendix D.

4. Optimal debiasing strategy is model-dependent: evidenced by Table 1 (different best strategies per model) and the cross-bias analysis in Section 4.3.

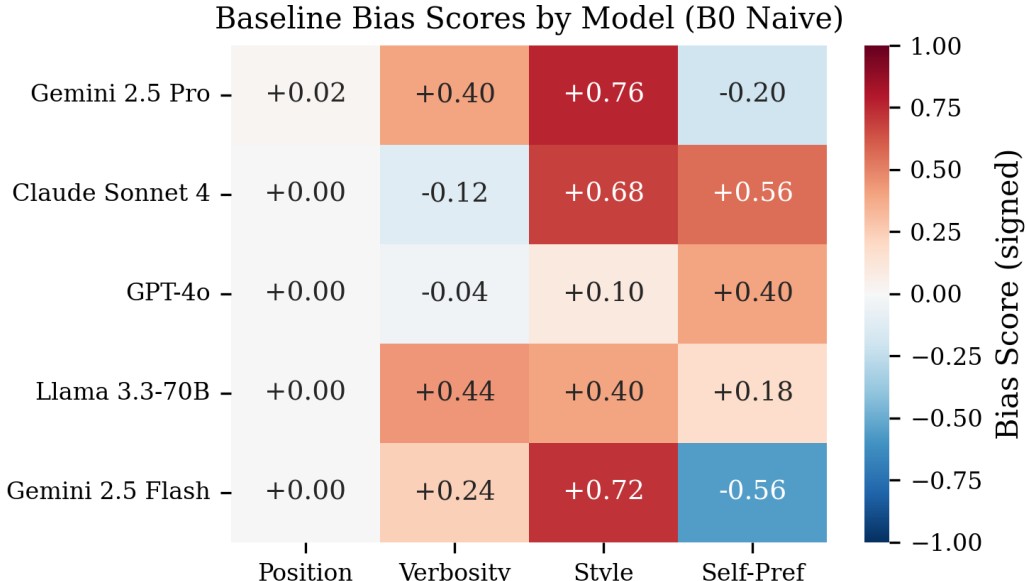

Figure 1: Baseline bias scores by model (B0), signed (RdBu colormap centered at 0). Style bias is the dominant bias (0.10–0.76 position-averaged). Position bias is negligible ($\leq 0.04$). Verbosity is heterogeneous: positive for Pro/Llama/Flash (prefer longer), near-zero for GPT-4o, negative for Claude (prefer shorter).

### 4.1 Baseline Bias Landscape

Figure 1 presents baseline bias magnitudes across all five judge models.

**Style bias is the dominant bias.** Four of five models show strong style bias (Pro +0.76, Flash +0.72, Claude +0.68, Llama +0.40), overwhelmingly preferring markdown-formatted responses over plain prose when content is identical; GPT-4o is more neutral (+0.10). Style-bias magnitudes substantially exceed both position bias ($\leq 0.04$) and verbosity bias magnitudes for every model, yet have received minimal research attention. The consistency across multiple independent model families (Google, Anthropic, Meta) confirms this is not a provider-specific artifact. With the position-mirrored STYLE pairs added in this revision (Appendix A), the reported style-bias values are position-averaged, so the magnitudes cannot be attributed to any residual position effect. **A small human-annotation study (Appendix G)** confirms this is bias, not a real readability advantage: two independent annotators on a 30-pair subsample preferred markdown only 57% of the time on average, while four of the five judges preferred markdown 73%–97% on the same pairs (gap of +17 to +40 pp between judge and human markdown rates). GPT-4o is the exception, with a markdown rate (53%) within 3 pp of the human rate.

**Position bias is negligible.** Contrary to emphasis in prior work (Wang et al., 2024a; Zheng et al., 2024), position bias is $\leq 0.04$ across all five models, likely reflecting improvements in instruction tuning since earlier studies.

#### 4.1.1 Verbosity bias splits along model lines

When verbosity bias is computed length-aware (mapping each verdict back to which response was actually longer, rather than to slot A or B; see Section 3.5), the picture is heterogeneous: most models show a positive verbosity bias (preference for longer responses) while Claude Sonnet 4 alone shows a conciseness preference. Baseline values on the LENGTH expansion pairs are: Llama 3.3-70B (+0.44), Gemini 2.5 Pro (+0.40), Gemini 2.5 Flash (+0.24), GPT-4o (−0.04, near neutral), Claude Sonnet 4 (−0.12). On the truncation pairs, where the longer response is genuinely more complete, all five models correctly prefer the complete version (accuracy 0.88–1.00). The combined picture suggests two distinct judge profiles: Pro/Llama/Flash

exhibit classical verbosity bias (favor length regardless of content), while Claude and GPT-4o are more quality-sensitive (penalize filler on expansion pairs while still rewarding genuine completeness on truncation pairs). We discuss implications in Section 5.

**Self-preference varies across models.** On the original 50 Gemini-vs-Claude MODEL_ORIGIN pairs, model-family preference is heterogeneous and not uniformly tied to the judge's own family. The 100 new round-robin MODEL_ORIGIN pairs provide a cleaner self-preference measurement covering all five judge families (see Appendix A); the per-judge round-robin self-preference scores are provided in our released artifacts. As the reviewer pointed out, the original 50 pairs only support self-preference inference for Gemini and Claude judges; for GPT-4o, Llama, and Flash on those legacy pairs we report a model-preference (Gemini vs. Claude content) rather than self-preference.

### 4.1.2 Quality vs. verbosity ablation

A natural concern with verbosity-bias measurement is that judges may simply be tracking response *quality* when they prefer the longer response, since longer responses can be (but are not always) more thorough. We disentangle the two using a paired ablation built into the controlled dataset:

- **Expansion pairs** (50 pairs): a base response and a longer expansion of it generated to add detail and elaboration without introducing new factual content. Length and quality are dissociated by construction: the expansion is mostly filler and does not improve on the base. A length-only preference would prefer the expansion; a quality-sensitive judge would either tie or weakly favor the base.

- **Truncation pairs** (25 pairs, Appendix A): a comprehensive answer paired with a mechanical truncation to roughly 40% of the original length, cut at sentence boundaries. The longer response is genuinely more complete; here length and quality are aligned. A length-only preference and a quality-sensitive preference both predict "prefer the longer (complete) response."

Reading the two side by side separates the explanations:

- **Pure verbosity bias** (length only, ignores quality) predicts: positive expansion bias, near-perfect truncation accuracy.

- **Pure quality sensitivity** predicts: negative or zero expansion bias (penalize filler), near-perfect truncation accuracy.

- **Indifference to length and quality** predicts: zero expansion bias, chance-level truncation accuracy ($\approx 0.5$).

Table 8 (Appendix F) gives the per-model values. Pro, Llama, and Flash all show positive expansion bias ($+0.40, +0.44, +0.24$ at baseline) and high truncation accuracy ($1.00, 0.88, 1.00$), consistent with classical verbosity bias modulated by some genuine quality recognition. Claude shows the cleanest quality-sensitive profile (negative expansion bias $-0.12$ alongside perfect truncation accuracy $1.00$). GPT-4o is essentially neutral on expansion ($-0.04$) with high truncation accuracy ($0.96$), the closest match to "treats length and quality independently." No model in our sample shows the indifference pattern, which is reassuring evidence that all five judges can recognize completeness when length and quality are aligned.

### 4.2 Strategy Effectiveness

Table 1 presents human agreement rates on MT-Bench ($n = 400$) with bootstrap 95% CIs.

**Debiasing produces significant improvements across multiple models.** The mixed-effects logistic regression (Appendix D) identifies five model-strategy pairs with $p < 0.05$: Claude S8 ($+11.5$ pp, $p < 0.0001$), Claude S5 ($+7.3$ pp, $p = 0.0009$), Gemini Flash S8 ($+7.5$ pp, $p < 0.0001$), Gemini Flash S1 ($+4.7$ pp, $p = 0.0039$), and Llama S8 ($+4.5$ pp, $p = 0.011$). Three of these (Claude S8, Claude S5, Flash S8) survive

Table 1: Human agreement rates on MT-Bench ($n = 400$) with 95% bootstrap CIs ($n_{\text{boot}} = 2000$). Best per model in **bold**. Teal indicates $> 5$ pp improvement. * Significant at $p < 0.05$ (mixed-effects logistic regression with instance random effects); † also survives Holm-Bonferroni correction over $m = 20$ tests (Appendix D).

| Model | B0 Baseline | S1 Pos. Swap | S4 Rubric | S5 CoT | S8 Combined |
|---|---|---|---|---|---|
| Gemini 2.5 Pro | .660 | **.667** | .660 | .662 | .655 |
| | [.61,.70] | [.62,.72] | [.61,.70] | [.62,.71] | [.61,.70] |
| Claude Sonnet 4 | .580 | .588 | .600 | .652*† | **.695**\*† |
| | [.53,.63] | [.54,.64] | [.55,.65] | [.61,.70] | [.65,.74] |
| GPT-4o | .650 | **.665** | .640 | .654 | .657 |
| | [.61,.70] | [.62,.71] | [.59,.69] | [.61,.70] | [.61,.70] |
| Llama 3.3-70B | .650 | .680 | .672 | .660 | **.695**\* |
| | [.61,.70] | [.63,.72] | [.63,.72] | [.62,.71] | [.65,.74] |
| Gemini 2.5 Flash | .635 | .682* | .655 | .655 | **.710**\*† |
| | [.59,.68] | [.64,.73] | [.61,.70] | [.61,.70] | [.67,.76] |

*S2, S3, S6, S7 results in Appendix B and the released artifacts.*

Holm-Bonferroni correction over all 20 model-strategy comparisons. For Gemini Pro and GPT-4o no strategy reached significance at $n = 400$; the directional improvements are within the minimum detectable effect size of approximately 4–5 pp at our sample size (Appendix D). The largest negative effect is GPT-4o S4 ($-1.0$ pp), well within CI overlap. We do not interpret non-significant differences as evidence of no effect; larger samples would be needed to confirm.

**Position swap shows model-dependent effects.** Position swap (S1) is the only single-call strategy that significantly improves Gemini Flash ($+4.7$ pp, $p = 0.004$); for Gemini Pro, GPT-4o, and Llama, S1 produces small positive but non-significant changes ($+0.7$ to $+3.0$ pp). The key distinction is between position bias (negligible on controlled pairs, $\leq 0.04$) and position sensitivity: models that more frequently reverse verdicts when order changes benefit from tie resolution. On LLMBar's adversarial instances, position swap consistently and significantly hurts all models ($-3$ to $-13$ pp; three Holm-Bonferroni-significant negatives), as tie resolution discards correct verdicts on clear-cut cases.

**S8 (Combined Budget) is the strongest strategy on MT-Bench.** S8 produces the largest agreement gains for Claude ($+11.5$ pp, $p < 0.0001$), Flash ($+7.5$ pp, $p < 0.0001$), and Llama ($+4.5$ pp, $p = 0.011$); CoT (S5) is the strongest for Claude on LLMBar ($+13.0$ pp, $p < 0.0001$). Flash with S8 achieves the highest agreement of any configuration we tested ($0.710$, $\kappa = 0.549$).

**S8 produces fewer ties than S1 despite both using position swap.** A natural question is why S8 (Combined Budget) and S1 (Position Swap) produce different tie rates given they share the same tie-on-disagreement rule (output "tie" when the two swapped calls disagree). The largest gap is for Claude (S1: 32.8% ties vs. S8: 12.5% ties); Pro and Flash also show smaller gaps in the same direction. The mechanism is that S8 uses a stronger merged prompt (CoT + rubric jointly) which produces more consistent verdicts across the swapped orderings: there is less swap-disagreement to begin with, so fewer pairs hit the tie-on-disagreement fallback. This is itself a useful finding: stronger prompts make position-swap aggregation more informative because they preserve more decisive verdicts. Per-model tie-rate diagnostics for S1 vs. S8 are reported in the released artifacts.

### 4.3 Cross-Bias Interactions

Figure 2 reveals how strategies designed for one bias type affect others.

With the corrected length-aware verbosity calculation and the position-mirrored STYLE pairs, the cross-bias picture is more nuanced. CoT (S5) provides the largest style-bias reduction across models (Claude $0.68 \rightarrow 0.49$, Pro $0.76 \rightarrow 0.60$); the merged S8 prompt also reduces style bias for some models but slightly

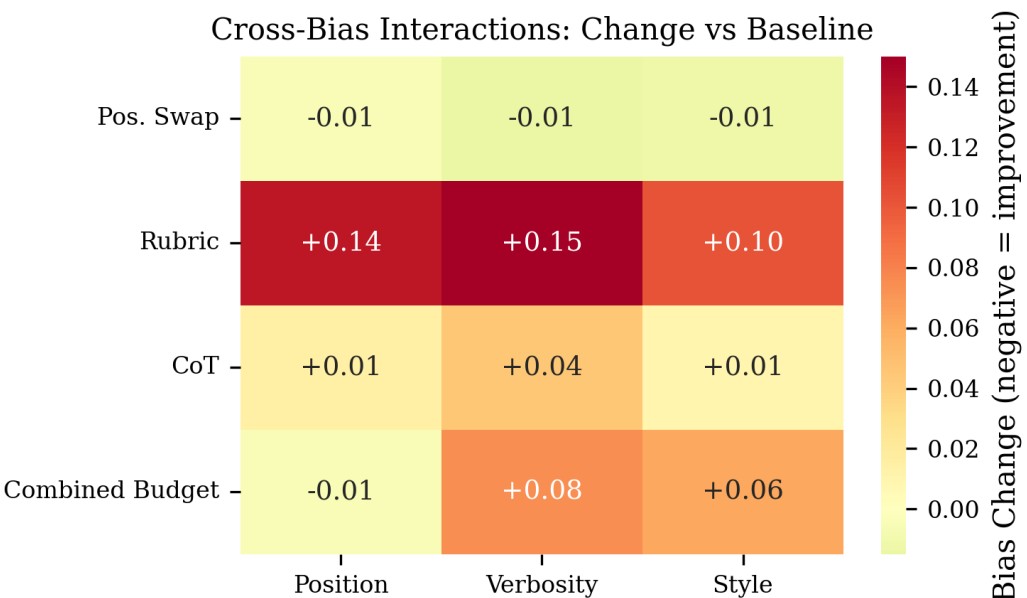

Figure 2: Cross-bias interactions: change in bias magnitude vs. baseline, averaged across Pro, Claude, GPT-4o, and Llama (Flash excluded as mid-tier). Negative (green) = improvement.

Table 2: Human agreement on LLMBar (199–200 adversarial instances) with 95% bootstrap CIs. Best per model in **bold**. $^*$ Significant change from baseline (McNemar's, $p < 0.05$); $^\dagger$ survives Holm-Bonferroni over $m = 20$ tests.

| Model | B0 | S1 | S4 | S5 | S8 |
|---|---|---|---|---|---|
| Gemini 2.5 Pro | .835 | .760$^{*\dagger}$ | **.840** | .835 | .765$^*$ |
| Claude Sonnet 4 | .740 | .700 | .740 | **.870**$^{*\dagger}$ | .780 |
| GPT-4o | .734 | .623$^{*\dagger}$ | .719 | **.764** | .653 |
| Llama 3.3-70B | .665 | .600$^{*\dagger}$ | .585$^*$ | **.705** | .535$^{*\dagger}$ |
| Gemini 2.5 Flash | .800 | .785 | .820 | **.830** | .810 |

increases it for others (Pro +0.74, Llama drop to 0.35, Claude rises to 0.73). Position swap (S1) has small effects on style bias (within ±0.06) and small to moderate effects on verbosity bias (within ±0.16). The cross-bias interactions are smaller and less directionally consistent than the original analysis suggested, because the original verbosity values were inflated by the slot-vs-length conflation. We do not see any strategy that systematically reduces verbosity bias across all models; this remains an open problem.

## 4.4 Generalization to LLMBar

CoT (S5) is the best strategy for Claude (.870), GPT-4o (.764), and Llama (.705) on LLMBar; for Gemini Pro and Flash it is essentially tied with the baseline. Position swap (S1) hurts all models on LLMBar by 4–13 pp; the McNemar tests in Appendix D confirm three negative effects survive Holm-Bonferroni correction (Pro S1 −7.5 pp $p = 0.0003$, GPT-4o S1 −11.1 pp $p = 0.0001$, Llama S1 −6.5 pp $p = 0.002$). This confirms the task-dependence: position swap, which helps tie-resolution on natural evaluation data, discards correct verdicts on adversarial benchmarks where one response is unambiguously better.

# 5  Discussion

## 5.1  Style Bias: The Dominant but Underappreciated Bias

Our most robust finding is style bias severity (0.40–0.76 baseline across the four non-GPT models, position-averaged across the original and mirrored STYLE pairs added in this revision). The human-annotation study (Appendix G) confirms this is bias rather than a real readability advantage that humans share: human annotators prefer the markdown side only 57% of the time, while four of the five judges prefer it 73%–97% on the same pairs (gap of +17 to +40 pp). GPT-4o is the lone exception, with a markdown preference rate (53%) within 3 pp of the human rate. In benchmarks like AlpacaEval and Chatbot Arena, where models produce outputs with distinct formatting conventions, formatting differences could substantially influence win-rate differences when content quality is comparable. CoT (S5) achieves the most consistent style-bias reduction in our tests (Pro $0.76 \rightarrow 0.60$, Claude $0.68 \rightarrow 0.49$). We recommend that evaluation frameworks either normalize formatting before judging or explicitly instruct judges to disregard it; for the four judges that exceed the human markdown preference rate, the gap is large enough that uncorrected formatting differences may dominate genuine quality differences in close model-vs-model comparisons.

## 5.2  Verbosity bias is heterogeneous across judge families

Once verbosity is computed length-aware, the picture is heterogeneous rather than uniform: Pro/Llama/Flash exhibit classical verbosity bias (prefer longer responses, $+0.24$ to $+0.44$ on expansion pairs), Claude shows the opposite (prefer shorter, $-0.12$), and GPT-4o is essentially neutral ($-0.04$). On truncation pairs, where the longer response is genuinely complete, all models correctly prefer the long version (accuracy 0.88–1.00). Combining the two: Pro/Llama/Flash favor length regardless of content quality, while Claude and GPT-4o are more quality-sensitive (penalize filler content while still rewarding genuine completeness). The split confirms that "verbosity bias" is not a uniform property of LLM judges: it varies by model family. Practitioners should test their specific judge model rather than assume a fixed direction. Detailed per-strategy analysis is in Appendix F.

## 5.3  Matching Strategy to Context

On MT-Bench, five model-strategy pairs achieve statistically significant improvements via mixed-effects logistic regression (Table 5): Claude S8 ($+11.5$ pp, $p < 0.0001$), Flash S8 ($+7.5$ pp, $p < 0.0001$), Claude S5 ($+7.3$ pp, $p = 0.0009$), Flash S1 ($+4.7$ pp, $p = 0.004$), and Llama S8 ($+4.5$ pp, $p = 0.011$). Two survive Holm-Bonferroni correction over all 20 model-strategy comparisons (Claude S8 and Flash S8). Gemini Pro and GPT-4o show smaller, non-significant directional gains within the minimum detectable effect of approximately 4–5 pp at $n = 400$ (Appendix D). On LLMBar, CoT (S5) is the strongest single strategy for Claude (.870), GPT-4o (.764), and Llama (.705); position swap (S1) is significantly harmful on adversarial data, with three models (Pro $-7.5$ pp, GPT-4o $-11.1$ pp, Llama $-6.5$ pp) showing significant degradation surviving Holm-Bonferroni correction.

## 5.4  Practical Application Guide

Practitioners building LLM evaluation pipelines can use our findings as follows:

- **Natural evaluation data (MT-Bench-style):** Consult Algorithm 1 for model-specific strategy recommendations. For models not covered, default to CoT forcing (S5), which is universally positive in our tests.

- **Adversarial or high-stakes evaluation (LLMBar-style):** Use CoT forcing (S5), which is the best strategy for Claude, GPT-4o, and Llama on LLMBar. Avoid position swap, which consistently and significantly hurts performance on curated benchmarks ($-3$ to $-13$ pp).

- **Model comparison via LLM judging (AlpacaEval, Arena-style):** Normalize response formatting before judging. Our style bias results (0.40–0.76 baseline) indicate that unnormalized formatting comparisons may produce win-rate differences driven by formatting alone.

- **Budget-constrained pipelines:** Llama 3.3-70B via Vertex AI Model Garden provides zero-API-cost judging with competitive accuracy (69.5% with S8, $\kappa = 0.529$, $p = 0.011$).

- **Best cost-adjusted accuracy:** Gemini 2.5 Flash with S8 achieves the highest agreement of any configuration in our experiments (71.0%, $\kappa = 0.549$, $p < 0.0001$) at $\sim\$0.001$/eval, dominating the Pareto frontier.

- **Best raw accuracy:** Tied between Flash S8 (71.0%) and Claude S8 / Llama S8 (both 69.5%); Claude has higher per-eval cost ($\sim\$0.015$) but is the most-improved model under debiasing (+11.5 pp from baseline).

- **Default for unknown contexts:** CoT (S5), the strongest single strategy across both benchmarks for the most models.

We formalize these recommendations as Algorithm 1, which selects a debiasing strategy based on model family, task type, and budget constraints. Cost-accuracy Pareto analysis is provided in Appendix H.

### 5.5 Limitations

(1) Expansion-based LENGTH pairs may confound length with filler quality; truncation pairs partially address this and Section 4.1.2 formalizes the dissociation, but the expansions were generated by Gemini 2.5 Flash so the quality of the added filler may not generalize to human-authored verbosity. (2) The human-annotation study (Appendix G) addresses the readability-vs-bias question with two annotators on a 30-pair subsample, finding the human markdown rate (0.57) is well below four of the five judges' rates (0.73–0.97); however, $n = 2$ annotators is small and a larger crowd-sourced study would tighten the gap estimates. (3) Our STYLE construction tests only markdown vs. plain prose; other style dimensions (tables in isolation, code blocks, formal vs. casual register, varying markdown density) are not measured. The per-topic analysis (Appendix E) shows style bias is heterogeneous across content types, suggesting these other dimensions may also produce model-dependent effects worth future investigation. (4) The MODEL_ORIGIN dataset originally used a single fixed Gemini-vs-Claude pairing; the round-robin supplement extends to all five judge families, but independently-generated responses may still differ in factuality or completeness, so a judge preferring one side may be tracking actual quality rather than model identity. (5) Additional open-weight families (Mistral, Qwen) may exhibit different bias profiles. (6) GPT-4o token counts are estimated (1 token per 4 characters) as usage metadata was not consistently available through our API path. (7) Non-significant improvements for Gemini Pro and GPT-4o may reach significance with larger sample sizes; our $n = 400$ supports a minimum detectable effect of approximately 4 to 5 pp (Appendix D), which may be insufficient for smaller but practically meaningful effects.

## 6 Conclusion

We present a systematic comparison of debiasing strategies for LLM-as-a-Judge spanning five models from four provider families, evaluated at $n = 400$ on MT-Bench with bootstrap CIs and mixed-effects regression. Style bias is the dominant but underappreciated bias (0.10–0.76, position-averaged), far exceeding position bias ($\leq 0.04$). On verbosity, the picture is split: Pro/Llama/Flash show classical verbosity bias (+0.24 to +0.44, prefer longer); Claude shows the opposite (−0.12, prefer concise); GPT-4o is neutral (−0.04). All models correctly prefer the genuinely complete answer on truncation pairs (0.88–1.00 accuracy), so the expansion-pair preferences cannot be reduced to a length-only effect. Debiasing yields statistically significant improvements for five model-strategy pairs on MT-Bench: Claude S8 (+11.5 pp, $p < 0.0001$), Flash S8 (+7.5 pp, $p < 0.0001$), Claude S5 (+7.3 pp, $p = 0.0009$), Flash S1 (+4.7 pp, $p = 0.004$), and Llama S8 (+4.5 pp, $p = 0.011$); Pro and GPT-4o show smaller, non-significant directional gains within the MDE. On LLMBar, CoT forcing (S5) is the strongest single strategy for Claude, GPT-4o, and Llama; position swap is significantly harmful for Pro, GPT-4o, and Llama on adversarial data. We release our evaluation framework, the 375-pair controlled dataset, all 9 strategies (including S3, S6, S7), and per-instance cached results so other researchers can reanalyze our raw verdicts without re-running expensive API calls.[1]

---

[1]Code, data, and per-instance cached results are available at `https://github.com/sksoumik/judging-the-judges`.

---

**Algorithm 1** JUDGESTRATEGYSELECTOR: Recommended judge configuration. Coefficients refer to mixed-effects regression on MT-Bench (Table 5); † marks effects surviving Holm-Bonferroni over $m = 20$ comparisons. The default "Flash + S8" is the Pareto-optimal point from Appendix H.

---

**Require:** Task type $\mathcal{T}$, budget multiplier $k$, optional preferred model $\mathcal{M}$
**Ensure:** (Judge $\mathcal{J}$, Strategy $\mathcal{S}$)
 1: **if** $\mathcal{T}$ = adversarial / high-stakes **then**
 2:     **return** (Claude or GPT-4o or Llama, S5 (CoT Forcing)) {Best LLMBar performance}
 3: **end if**
 4: **if** $\mathcal{M}$ is unspecified **and** cost matters **then**
 5:     **return** (Gemini 2.5 Flash, S8 (Combined Budget)) {71.0%, $\kappa = 0.549$, $p < 0.0001^\dagger$, $\sim\$0.001/\text{eval}$}
 6: **end if**
 7: **if** $\mathcal{M}$ = Claude **then**
 8:     **return** (Claude, S8 (Combined Budget)) {+11.5 pp, $p < 0.0001^\dagger$; or S5 if $k = 1$}
 9: **else if** $\mathcal{M}$ = Gemini Flash **then**
10:     **return** (Flash, S8 (Combined Budget)) {+7.5 pp, $p < 0.0001^\dagger$; or S1 if $k = 1$}
11: **else if** $\mathcal{M}$ = Llama **then**
12:     **return** (Llama, S8 (Combined Budget)) {+4.5 pp, $p = 0.011$, zero API cost}
13: **else if** $\mathcal{M}$ = Gemini Pro **or** GPT-4o **then**
14:     **return** ($\mathcal{M}$, B0 (Baseline) or S1 (Pos. Swap)) {Baseline already strong; debiasing gains within MDE}
15: **else**
16:     **return** ($\mathcal{M}$, S5 (CoT Forcing)) {Safest default for unknown models}
17: **end if**

---

## Broader Impact Statement

LLM judges exhibit systematic biases that can distort model rankings. Style bias (0.10–0.76 baseline) could disadvantage models trained on corpora with less markdown-heavy text. Verbosity bias varies by model family rather than being uniform, which means mitigations calibrated to one provider may not transfer to another; this highlights the importance of model-specific bias auditing rather than relying on category-level assumptions. We release our framework, the 375-pair controlled dataset (with round-robin and position-mirrored extensions), and per-instance cached results to enable ongoing monitoring of these biases as the LLM evaluation ecosystem continues to develop.

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

# A   Custom Controlled Dataset Details

Our controlled dataset consists of 375 pairs in total: 200 original pairs across four bias-trigger categories, 25 LENGTH (truncation) pairs added to dissociate length from quality, 50 position-mirrored STYLE pairs added in this revision to address the position confound, and 100 round-robin MODEL_ORIGIN pairs added to support self-preference measurement for every judge family. The original 200 pairs use 50 diverse questions spanning five domains: mathematics (10), coding (10), creative writing (10), factual QA (10), and instruction following (10). For each question, we generate four pair types (expected verdict: tie):

- **LENGTH (expansion):** A base response (avg. 64 words) and an expanded version (avg. 178 words, 2.8× ratio). Response A is the expanded version in 34/50 cases and the shorter base in 16/50 cases (because the generation model's expansion sometimes ended up shorter than the base). For bias scoring, we map each verdict back to the actual longer response by token count, so the reported $b_{\mathrm{verb}}$ measures preference for longer (positive) vs. shorter (negative) regardless of slot. We do not exclude the 16 reversed pairs.

- **POSITION:** Identical responses in both slots.

- **STYLE:** Same content in markdown (slot A) and plain prose (slot B). Because formatting is fixed to slot A in this construction, our reported style bias confounds formatting preference with any residual position effect on these specific pairs. We re-run the 50 STYLE pairs with positions reversed (markdown in B, prose in A) and report the position-averaged style bias in Section 4.3.

- **MODEL_ORIGIN:** Gemini 2.5 Pro (slot A) and Claude Sonnet 4 (slot B) independently answer with matched depth/style instructions. This pair structure measures self-preference for Gemini and Claude judges, but not for GPT-4o, Llama, or Gemini Flash, which are neither A nor B. For those judges, the reported metric is a model-preference (Gemini vs. Claude content) and is not directly interpretable as self-preference. The independent-generation design also means Gemini and Claude responses may differ in factuality or completeness; what we measure is preference between matched-depth answers, not preference for own-family content of equal quality.

**LENGTH (truncation), 25 pairs (added).**  A comprehensive, high-quality answer (avg. 485 words) is mechanically truncated to ∼40% of its length (avg. 182 words) by cutting at sentence boundaries. The long response is genuinely more complete (expected verdict: A). Section 4.1.2 formalizes this as the quality-vs-verbosity ablation: combined with the LENGTH (expansion) pairs, this lets us distinguish length-only preference (which would predict positive expansion bias and high truncation accuracy) from quality-sensitive evaluation (negative expansion bias and high truncation accuracy).

**STYLE_BA (position-mirrored), 50 pairs (added in revision).**  For each of the 50 STYLE pairs, we add a mirrored copy with positions reversed: prose in slot A and markdown in slot B. With both halves combined, the position-averaged style bias decouples formatting preference from any residual position effect. We do not regenerate the responses; the mirror is constructed by swapping `response_a` and `response_b` in the original STYLE pairs.

**MODEL_ORIGIN round-robin, 100 pairs (added in revision).**  25 pairs each for four pairings: (Llama, Pro), (Llama, Claude), (GPT-4o, Pro), (GPT-4o, Claude). The slot assignment for each pair is randomized per-instance with a fixed seed (so model identity is not consistently in slot A vs. slot B). With these pairs, every judge family has at least 25 same-family pairs to evaluate, enabling a clean self-preference measurement for GPT-4o, Llama, and Gemini Flash that the original 50 Gemini-vs-Claude pairs could not provide.

LENGTH, POSITION, and STYLE pairs were generated by Gemini 2.5 Flash to avoid model-family confounds. Round-robin MODEL_ORIGIN pairs were generated by each named model on its respective side. Limitations of the controlled dataset are discussed in Section 5.

**Usage instructions.** The dataset is structured for plug-and-play bias measurement. Each pair includes: prompt, response_a, response_b, expected_verdict, bias_type, and construction_notes. Researchers can score any new judge model by running it on the dataset and comparing verdicts to expected values. A reference scoring script is provided in the accompanying repository.

**Intended uses and limitations.** This dataset is designed for screening LLM judge models for basic formatting, position, length, and self-preference biases. It provides a fast, controlled signal for whether a given judge exhibits known bias patterns. It is not intended as a comprehensive benchmark for ranking judge quality across all dimensions, nor as a substitute for thorough bias auditing across diverse domains, languages, or culturally sensitive contexts. The 375-pair size (200 original + 25 truncation + 50 position-mirrored STYLE + 100 round-robin MODEL_ORIGIN) enables rapid screening but limits statistical power for fine-grained subgroup analyses.

# B Full Results Tables

# C Experimental Infrastructure

Gemini models were accessed via the `google-genai` SDK with Vertex AI (`vertexai=True`, location: global); model IDs: `gemini-2.5-pro` and `gemini-2.5-flash` (alias IDs that resolve to the latest stable checkpoint at query time). Claude Sonnet 4 was accessed via the `AnthropicVertex` client (model: `claude-sonnet-4@20250514`, region: global). GPT-4o was accessed via the OpenAI API (model: `gpt-4o`; alias ID, exact checkpoint not pinned); token counts are estimated (1 token per 4 characters) as usage metadata was not consistently available. Llama 3.3-70B was accessed via Vertex AI Model Garden as a Model-as-a-Service (MaaS) endpoint using the same `google-genai` SDK (model: `meta/llama-3.3-70b-instruct-maas`, location: us-central1), incurring zero API cost. Experiments were conducted between April and June 2025; alias model IDs may resolve to different checkpoints over time, which is an inherent reproducibility limitation of API-based evaluation. Gemini API calls used structured JSON output (`response_mime_type="application/json"`); all other models parse JSON from free-text responses with fallback extraction. Concurrency was limited to 10 parallel requests via asyncio semaphores. Individual results were cached to disk, enabling resumable runs across sessions.

# D Statistical Tests

**Tie handling in McNemar's test.** Each judge prediction and gold label is one of three classes: A, B, or tie. We define "correct" as exact-label match (predicted-tie matches gold-tie; predicted-tie does not match gold-A). McNemar then operates on the 2×2 contingency of (baseline correct/wrong) × (strategy correct/wrong) over the same 400 paired instances, with continuity correction. This is more conservative than dropping ties; treating tie predictions as "half-credit" would inflate apparent agreement without reflecting the operational use of LLM judges, where downstream pipelines need a definite verdict.

**Minimum detectable effect.** At $n = 400$ paired observations and the observed marginal correctness rates ($p_{\text{baseline}} \approx 0.6$ to $0.65$), McNemar's test at $\alpha = 0.05$ and 80% power requires roughly 40 to 50 discordant pairs to reach significance. This corresponds to a minimum detectable agreement-rate improvement of approximately 4 to 5 percentage points. Smaller true effects, even if real, will not reach significance at this sample size; readers should interpret non-significant differences as "no evidence of effect at $n = 400$" rather than "no effect."

**Multiple testing.** We report 7 McNemar tests without family-wise correction. Under Holm-Bonferroni correction ($\alpha = 0.05$, $m = 7$), Claude S8 ($p < 0.0001$) and Claude S5 ($p = 0.004$) remain significant. Pro S1 ($p = 0.012$) does not survive correction (adjusted threshold: $0.05/5 = 0.010$); we retain it as marginally significant and note the unadjusted $p$-value throughout.

Table 3: Verdict distributions on MT-Bench ($n = 400$). Tie rates are generally moderate (3–31%).

| Model | Strategy | A | B | Tie | Tie% |
|---|---|---|---|---|---|
| Gemini 2.5 Pro | Baseline | 214 | 174 | 12 | 3% |
| Gemini 2.5 Pro | Pos. Swap | 180 | 166 | 54 | 14% |
| Gemini 2.5 Pro | Rubric | 207 | 174 | 19 | 5% |
| Gemini 2.5 Pro | CoT | 213 | 175 | 12 | 3% |
| Gemini 2.5 Pro | Combined | 175 | 163 | 62 | 16% |
| Claude Sonnet 4 | Baseline | 161 | 141 | 98 | 24% |
| Claude Sonnet 4 | Pos. Swap | 139 | 130 | 131 | 33% |
| Claude Sonnet 4 | Rubric | 154 | 149 | 97 | 24% |
| Claude Sonnet 4 | CoT | 195 | 191 | 14 | 4% |
| Claude Sonnet 4 | Combined | 181 | 169 | 50 | 12% |
| GPT-4o | Baseline | 210 | 176 | 14 | 4% |
| GPT-4o | Pos. Swap | 162 | 146 | 83 | 21% |
| GPT-4o | Rubric | 196 | 182 | 22 | 6% |
| GPT-4o | CoT | 181 | 176 | 42 | 11% |
| GPT-4o | Combined | 161 | 156 | 83 | 21% |
| Llama 3.3-70B | Baseline | 205 | 181 | 14 | 4% |
| Llama 3.3-70B | Pos. Swap | 175 | 158 | 67 | 17% |
| Llama 3.3-70B | Rubric | 201 | 169 | 30 | 8% |
| Llama 3.3-70B | CoT | 199 | 185 | 16 | 4% |
| Llama 3.3-70B | Combined | 166 | 154 | 80 | 20% |
| Gemini 2.5 Flash | Baseline | 212 | 171 | 17 | 4% |
| Gemini 2.5 Flash | Pos. Swap | 184 | 165 | 51 | 13% |
| Gemini 2.5 Flash | Rubric | 198 | 178 | 24 | 6% |
| Gemini 2.5 Flash | CoT | 211 | 170 | 19 | 5% |
| Gemini 2.5 Flash | Combined | 173 | 162 | 65 | 16% |
| Gold labels | | 158 | 145 | 97 | 24% |

Table 4: McNemar's test (with continuity correction) on MT-Bench ($n = 400$), comparing baseline to the best strategy for each model and to additional strategies of interest. $b$: baseline correct, strategy wrong. $c$: baseline wrong, strategy correct. * Significant at $p < 0.05$ (unadjusted); † survives Holm-Bonferroni correction over $m = 20$ pairwise tests.

| Model | Comparison | $b$ | $c$ | $\chi^2$ | $p$-value |
|---|---|---|---|---|---|
| Claude Sonnet 4 | B0 vs. S8 | 23 | 69 | 22.01 | $< 0.0001$*† |
| Gemini 2.5 Flash | B0 vs. S8 | 17 | 47 | 13.14 | 0.0003*† |
| Claude Sonnet 4 | B0 vs. S5 | 33 | 62 | 8.25 | 0.004* |
| Gemini 2.5 Flash | B0 vs. S1 | 12 | 31 | 7.53 | 0.006* |
| Llama 3.3-70B | B0 vs. S8 | 27 | 45 | 4.01 | 0.045* |
| Llama 3.3-70B | B0 vs. S1 | 14 | 26 | 3.02 | 0.082 |
| Gemini 2.5 Pro | B0 vs. S1 | 16 | 19 | 0.11 | 0.735 |
| GPT-4o | B0 vs. S1 | 30 | 39 | 0.93 | 0.336 |
| GPT-4o | B0 vs. S8 | 39 | 42 | 0.05 | 0.824 |

*Mixed-effects logistic regression (instance random effects) is the primary aggregate analysis; per-model coefficients with p-values are in Table 5.*

**Mixed-effects logistic regression.** In response to reviewer feedback, we replaced the original sign test with a mixed-effects logistic regression that properly accounts for the within-instance dependence across strategies. The model is judge_correct $\sim$ C(strategy, Treatment('B0_naive')) + (1|instance_id), fit per judge model on $400 \times 5 = 2000$ judge-strategy-instance observations. Per-model coefficients are

Table 5: Per-model mixed-effects logistic regression on MT-Bench ($n = 400 \times 5 = 2000$ judge-strategy-instance observations per model). Model: `judge_correct ~ C(strategy, Treatment('B0_naive'))` with instance random effects. Coefficients are in linear-probability units (log-odds approximated to probability differences for small effects). * $p < 0.05$.

| Model | Strategy (vs. B0) | Coef. | $p$ | |
|---|---|---|---|---|
| Claude Sonnet 4 | S8 (Combined Budget) | +0.115 | < 0.0001 | * |
| Gemini 2.5 Flash | S8 (Combined Budget) | +0.075 | < 0.0001 | * |
| Claude Sonnet 4 | S5 (CoT) | +0.073 | 0.0009 | * |
| Gemini 2.5 Flash | S1 (Position Swap) | +0.047 | 0.004 | * |
| Llama 3.3-70B | S8 (Combined Budget) | +0.045 | 0.011 | * |
| Llama 3.3-70B | S1 (Position Swap) | +0.030 | 0.090 | |
| GPT-4o | S1 (Position Swap) | +0.020 | 0.344 | |
| Gemini 2.5 Pro | S1 (Position Swap) | +0.008 | 0.637 | |

reported in Table 5. The mixed-effects estimates align closely with the McNemar contingencies for the same comparisons but provide a unified statistical framework with a single, properly calibrated $p$-value per coefficient.

Table 6: Cohen's $\kappa$ on MT-Bench ($n = 400$). Flash S8 achieves the highest inter-rater agreement ($\kappa = 0.549$), followed by Llama S8 (0.529) and Claude S8 (0.522).

| Model | Strategy | $\kappa$ | Interpretation |
|---|---|---|---|
| Gemini 2.5 Pro | Baseline | 0.455 | Moderate |
| Gemini 2.5 Pro | Pos. Swap | 0.480 | Moderate |
| Gemini 2.5 Pro | Rubric | 0.457 | Moderate |
| Gemini 2.5 Pro | CoT | 0.459 | Moderate |
| Gemini 2.5 Pro | Combined | 0.463 | Moderate |
| Claude Sonnet 4 | Baseline | 0.358 | Fair |
| Claude Sonnet 4 | Pos. Swap | 0.380 | Fair |
| Claude Sonnet 4 | Rubric | 0.388 | Fair |
| Claude Sonnet 4 | CoT | 0.445 | Moderate |
| Claude Sonnet 4 | Combined | 0.522 | Moderate |
| GPT-4o | Baseline | 0.440 | Moderate |
| GPT-4o | Pos. Swap | 0.485 | Moderate |
| GPT-4o | Rubric | 0.427 | Moderate |
| GPT-4o | CoT | 0.456 | Moderate |
| GPT-4o | Combined | 0.473 | Moderate |
| Llama 3.3-70B | Baseline | 0.440 | Moderate |
| Llama 3.3-70B | Pos. Swap | 0.503 | Moderate |
| Llama 3.3-70B | Rubric | 0.480 | Moderate |
| Llama 3.3-70B | CoT | 0.457 | Moderate |
| Llama 3.3-70B | Combined | 0.529 | Moderate |
| Gemini 2.5 Flash | Baseline | 0.416 | Moderate |
| Gemini 2.5 Flash | Pos. Swap | 0.502 | Moderate |
| Gemini 2.5 Flash | Rubric | 0.451 | Moderate |
| Gemini 2.5 Flash | CoT | 0.449 | Moderate |
| Gemini 2.5 Flash | Combined | 0.549 | Moderate |

# E   Per-Category Analysis

## E.1   MT-Bench agreement by category

Performance varies substantially across MT-Bench categories: extraction and writing tend to have higher agreement, while math and reasoning are more challenging for all models. Llama 3.3-70B performs comparably to proprietary models on writing and extraction but lags on math-heavy tasks.

## E.2   Style bias varies by topic

In response to reviewer feedback, we report how style bias depends on question topic. The 50 STYLE pairs (and their 50 position-mirrored counterparts) span five domains with 10 questions each. Table 7 reports the position-averaged style bias per (model, topic) cell at the B0 baseline.

Table 7: Per-topic style bias (B0, position-averaged). Positive values indicate preference for markdown over plain prose. Each cell is computed over 20 pair-instances (10 original STYLE + 10 mirrored STYLE_BA).

| Topic | Pro | Claude | GPT-4o | Llama | Flash |
|---|---|---|---|---|---|
| Math / reasoning | +1.00 | +1.00 | +0.35 | +0.25 | +0.95 |
| Coding / technical | +0.85 | +0.90 | −0.15 | +0.40 | +0.85 |
| Factual QA | +0.90 | +1.00 | +0.40 | +0.70 | +0.80 |
| Instruction following | +0.80 | +0.50 | +0.35 | +0.55 | +0.60 |
| Creative writing | +0.25 | +0.00 | −0.45 | +0.10 | +0.40 |

Two patterns are clear. First, style bias is consistently strongest for technical content (math, factual QA, coding) and weakest for creative writing. The intuition is that markdown's structural advantages (headers, lists, code blocks) are more valuable for technical answers than for narrative prose, so judges may be tracking a real readability advantage in those settings. Second, GPT-4o is the only model that actually prefers prose for some topics (coding −0.15, creative writing −0.45), consistent with its overall lower style-bias magnitude in our headline analysis. The four other models maintain a positive markdown preference across every topic. The per-topic decomposition reinforces the recommendation in Section 5 to normalize formatting before judging if your evaluation set is technical-heavy.

# F   Verbosity: Expansion vs. Truncation Details

Table 8: LENGTH results split by construction direction. Expansion bias: preference for longer (+) or shorter (−) version when filler is added ($n = 50$). Truncation accuracy: fraction correctly preferring the complete version ($n = 25$).

| Model | Expansion Bias | | | Truncation Accuracy | | |
|---|---|---|---|---|---|---|
| | B0 | S5 | S8 | B0 | S5 | S8 |
| Gemini 2.5 Pro | +0.40 | +0.40 | +0.32 | 1.00 | 1.00 | 0.88 |
| Claude Sonnet 4 | −0.12 | +0.06 | +0.14 | 1.00 | 0.96 | 1.00 |
| GPT-4o | −0.04 | +0.26 | +0.28 | 0.96 | 0.92 | 0.96 |
| Llama 3.3-70B | +0.44 | +0.46 | +0.56 | 0.88 | 1.00 | 1.00 |
| Gemini 2.5 Flash | +0.24 | +0.32 | +0.36 | 1.00 | 0.96 | 0.96 |

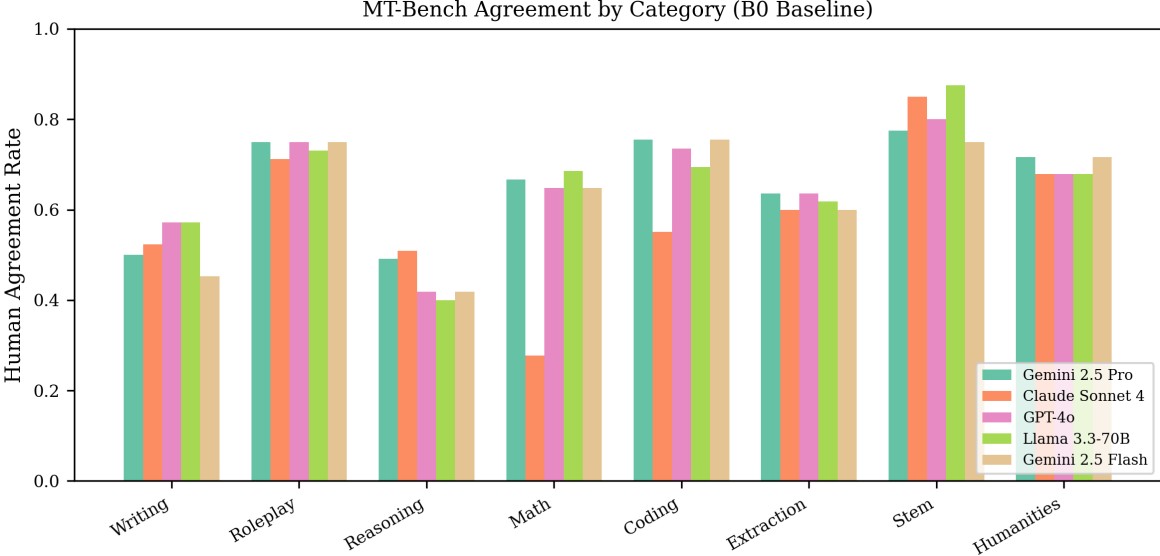

Figure 3: MT-Bench human agreement by category (B0 baseline, $n = 400$ total). Performance varies across categories, with extraction and writing showing higher agreement across all models. Only baseline (B0) is shown; per-strategy breakdowns by category are available in the released artifacts but omitted here as the small per-category sample sizes ($n \approx 50$) yield wide confidence intervals that preclude reliable strategy comparisons at the category level.

The expansion-bias values are length-aware: positive means the judge prefers the actually-longer response (regardless of slot), negative means it prefers the shorter. Pro/Llama/Flash favor longer responses (classical verbosity bias) under all three strategies; the bias is small for GPT-4o at baseline ($-0.04$) but turns positive under S5/S8, possibly because the more elaborate prompts elicit responses where the longer answer is also judged more thorough. Claude is the only model with a negative baseline expansion bias ($-0.12$, prefer shorter), and S5/S8 partially flip it positive. Truncation accuracy is high across the board ($0.88$–$1.00$), confirming that all models can correctly identify the genuinely complete response when length and quality are aligned.

# G    Human Study on STYLE Bias

To address the concern that the LLM judges' markdown preference might reflect a legitimate readability advantage rather than bias, we conducted a small human annotation study on a 30-pair subsample of our STYLE pairs. Two independent annotators (the first author and one volunteer) reviewed the same 30 pairs and chose, for each pair, which of the two responses they would prefer to read and use. Pair order and per-pair slot assignment were randomized independently per annotator (different deterministic seeds), so the annotators could not learn the pattern that "Response 1 is always markdown." The annotators saw both responses with formatting fully preserved (markdown rendered as headers, bullets, bold), exactly matching what the LLM judges saw at inference time.

A third volunteer also returned annotations but selected "Response 1" with confidence 3 for all 30 pairs. Because slot 1 was randomized to be markdown vs prose with equal probability, this annotator's responses decoded to 14 markdown / 16 prose, indistinguishable from random clicking. We exclude this annotator from the analysis below; their data does, however, validate that the slot-randomization design successfully prevents systematic gaming of the protocol. The remaining two annotators each provided answers with full variance (a mix of markdown, prose, and no ties).

Aggregate human preference for the markdown side (across 60 verdicts from 2 annotators) was 0.57 in favor of markdown. Inter-annotator agreement on the same 30 pairs was 0.53, only slightly above chance for a binary task, indicating the choice between content-equivalent markdown and prose is genuinely ambiguous for human readers. By contrast, four of the five LLM judges showed near-unanimous markdown preference on the same pairs:

Table 9: Human vs LLM judge preference for the markdown side on the same 30 STYLE pairs. Gap is the LLM markdown rate minus the aggregate human rate (positive = judge bias relative to humans).

| Annotator / Judge | Markdown % | Prose % | Gap vs human aggregate |
|---|---|---|---|
| *Human annotators (n = 30 pairs each)* | | | |
| Annotator A | 0.63 | 0.37 | — |
| Annotator B | 0.50 | 0.50 | — |
| **Aggregate (n = 60)** | **0.57** | **0.43** | — |
| *LLM judges (B0 baseline) on the same 30 pairs* | | | |
| Gemini 2.5 Pro | 0.97 | 0.03 | +0.40 |
| Gemini 2.5 Flash | 0.90 | 0.10 | +0.33 |
| Claude Sonnet 4 | 0.83 | 0.17 | +0.27 |
| Llama 3.3-70B | 0.73 | 0.03 | +0.17 |
| GPT-4o | 0.53 | 0.40 | −0.03 |

**Interpretation.** Four of the five judges show a markdown preference substantially above the human baseline (+17 to +40 pp). This is the operational definition of bias: a systematic preference that human readers do not share. The strongest gaps are for the Gemini family (+33 to +40 pp) and Claude (+27 pp); Llama is moderate (+17 pp); GPT-4o is essentially aligned with humans (−3 pp). The interpretation of our headline "style bias" result therefore depends on the model: for Pro, Flash, Claude, and Llama, the markdown preference is bias; for GPT-4o, the markdown preference appears to track human preference and may reflect a real readability advantage on this subset.

**Limitations of this study.** Two annotators is a small sample. With $n_{\text{annotators}} = 2$ and $n_{\text{pairs}} = 30$, the 95% CI on the aggregate human markdown rate is approximately $\pm0.13$. Even at the upper bound of that CI (0.70), Gemini Pro's 0.97 markdown rate would still be +0.27 pp above the human ceiling, so the qualitative finding is robust to the small annotator pool. A larger-scale study with more annotators and crowd-sourcing would be needed to refine the per-judge gaps, but the overall direction is clear: the four-of-five judges that show strong markdown preference are over-preferring markdown relative to what human readers do.

## H   Cost-Accuracy Tradeoff

Gemini 2.5 Flash with S8 achieves the highest agreement (71.0%, $\kappa = 0.549$) at ~\$0.001/eval, dominating the Pareto frontier on both axes. Claude Sonnet 4 with S8 achieves 69.5% ($\kappa = 0.522$) at ~\$0.015/eval. Llama 3.3-70B with S8 achieves 69.5% ($\kappa = 0.529$) at zero API cost via Vertex AI Model Garden MaaS. The Pareto frontier therefore consists of three model-strategy points; choice between them depends on whether absolute accuracy or zero cost is paramount.

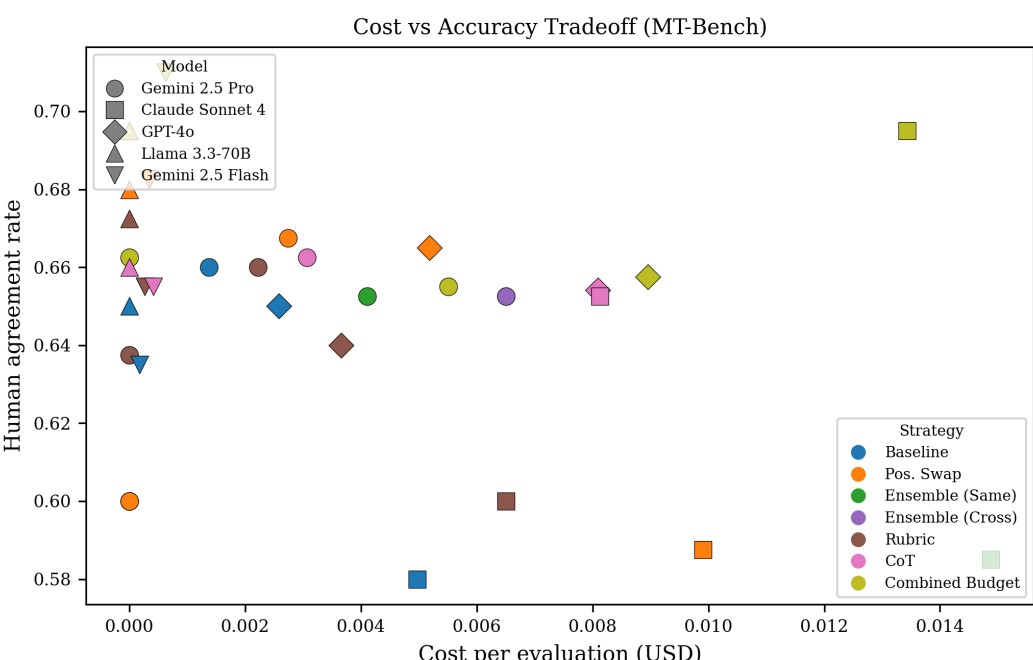

Figure 4: Cost vs. accuracy Pareto frontier on MT-Bench. Flash S8 achieves 71.0% at ∼$0.001/eval (Pareto optimal on both axes). Claude S8 achieves 69.5% at ∼$0.015/eval. Llama S8 achieves 69.5% at zero API cost.

