# OpenReview forum: "Judging the Judges: A Systematic Evaluation of Bias Mitigation Strategies in LLM-as-a-Judge Pipelines"
_TMLR — Accepted by TMLR_

### Review · Reviewer_GR4A · 2026-04-27

**Summary Of Contributions:**

This paper empirically compares nine debiasing strategies for LLM-as-a-Judge pipelines across five models, three benchmarks, and four bias types, claiming that (1) style bias dominates (other biases are significantly lower) , (2) models prefer conciseness over verbosity, (3) debiasing is beneficial but model-dependent, and (4) CoT is universally beneficial.  The topic is important and timely, but the submission is not sufficiently clear or rigorous in its present form.

*Strengths*

* Five models from four families, three benchmarks, nine strategies is the widest comparison in this space and is genuinely useful to the community.
* Bootstrap CIs throughout Table 1, McNemar tests with Holm-Bonferroni correction, and explicit acknowledgment of non-significant results for three of five models is good.
* The methodological separation of length-as-filler from length-as-completeness is the paper's cleanest contribution and a genuine conceptual advance over prior verbosity bias work.

*Major Weaknesses*
* Lack of clarity regarding what exactly each type of bias represent and how are they calculated.
* Lack of clarity on self-preference bias, from description it looks more like model family preference as its being judged by a different model (with different model family), rather than the one who generated it.
* Style bias calculated with positions fixed, confounding it with positional bias
* Outputs of two models are being scored, and bias is attributed when a judge prefers a response from a model, without considering if that actually produced better quality responses.

**Audience:**

Yes

**Audience Explanation:**

Yes. LLM-as-a-Judge is now quite important to how the ML community evaluates language models.
The finding that style bias is the dominant failure mode of current-generation judges is directly actionable and position bias is not that much of a problem is really useful. Similarly, other bias types discussions could be useful to the readers.

**Claims And Evidence:**

Yes

**Claims Explanation:**

*Weaknesses*
* The paper draws conclusions before discussing the results and also before describing how exactly do they measure the bias.
* Bias score is never defined per bias type.. Section 3.5 states that bias scores are computed as P(prefers A) − P(prefers B). This is the entirety of the formal definition. What exactly A and B is never specified for each of the four bias types in any unified location (except for position swap).
* For MODEL_ORIGIN pairs, they explicitly state that response A is always gemini and response B is always claude. They then proceed to report the self-preference scores for GPT  and Llama based on these pairs. Its totally unclear how is it self-preference. It only measures  third-party preference between two rival models.
* The strongest empirical claim, that style bias dominates all other biases, is also not cleanly supported. Appendix A states that STYLE pairs use markdown as response A and plain prose as response B. But it makes mixes the style and position. The paper’s position-bias control uses identical responses, but that does not rule out interaction between formatting and position.
* Appendix A says gemini  and claude independently answer the same prompts with matched depth/style instructions. This does not isolate self-preference or model origin. The outputs may differ in factuality, clarity, etc... How to know its not prefering a answer due to better quality rather than preference for some model or family? If Claude's responses are systematically better quality on these prompts, then a judge preferring response B is exhibiting correct evaluation behavior, not anti-self-preference or cross-preference.
* Appendix A also says that the expanded response is in slot A in 34/50 cases and slot B in 16/50 cases. So A is not necessarily longer response always. The bias score formula is not equivalent to P(prefers longer) − P(prefers shorter) for this bias type.
* For the 16/50 LENGTH pairs where expansion is in slot B, the formula would have the opposite sign. How are these cases handled? Are they excluded? Flipped?
* Fig 1 attempts to show the magnitude of the bias so reports the absolute value whereas section 4.1.1 reports the actual values. This is confusing and there should be clarification.
* In S1 (position swap), ties are declared on disagreement. Does S8 use the same tie-resolution rule? If so, Table 3 shows Claude S8 has 14% tie ratevs. Claude S1 at 31% ... dramatically fewer ties despite using position swap. Why?
* S4 (rubric) derives verdicts from summed scores across five criteria (1–10 each). S5 (CoT) derives verdicts from step-by-step reasoning. How are these merged?
* How 400 samples were selected?
* The judge outputs one of three verdicts: A, B, or tie. Section 3.5 does not explain how ties are handled in the McNemar table. Are ties excluded?
* Section 3.3 states that S3, S6, and S7 results are available in our released artifacts. These artifacts are unavailable... Ideally they should be present in the paper, along with the released resources.
* S1,S2,S3,..... does not carry any meaning, they are mostly unrelated to each other and makes it confusing which bias is under discussion. Better would be to stick with actual names

**Requested Changes:**

[Critical] Clearly define the bias score for each bias type in a single unified location. The current definition P(prefers A)−P(prefers B) is underspecified without explicitly stating what A and B correspond to for every bias category.

[Critical] Reorganize the paper so that methodological definitions precede any claims or conclusions. At present, conclusions are presented before the bias measurement framework is fully specified, which makes it difficult to interpret the results.

[Critical] Clarify the interpretation of “self-preference” in the MODEL_ORIGIN setting. As currently described, comparisons between fixed responses (e.g., Gemini as A and Claude as B) do not measure self-preference for other models such as GPT or LLaMA.

[Critical] Ensure that all bias measurements isolate the intended factor. In particular, the STYLE setup conflates formatting (markdown vs. prose) with positional effects, and the current position-bias control (identical responses) does not rule out interaction effects.

[Critical] Address the confounding between bias and response quality in MODEL_ORIGIN experiments. Since responses are independently generated, differences in factuality, clarity, or completeness can influence preferences.

[Critical] Resolve the inconsistency in the LENGTH bias formulation. When the longer response appears in different slots (A vs. B), the bias score as defined is not directly interpretable unless signs are consistently aligned. The paper must explicitly describe whether such cases are flipped, normalized, or excluded

[Major] Standardize the reporting of bias magnitudes across figures and text. The use of absolute values in Figure 1 and signed values in Section 4.1.1 creates confusion.

[Major] Provide a clear description of how verdicts from different evaluation strategies are combined. In particular, rubric-based and S5 CoT appear to produce different types of outputs. The paper should specify how these are converted into comparable A/B/tie decisions and whether any aggregation or normalization is applied.

[Major] Describe the data sampling process in detail. The paper currently does not explain how the 400 samples were selected, whether they are stratified across tasks or domains, or whether any filtering criteria were applied

[Major] Ensure that all referenced artifacts are publicly available and main observations and results are present in the paper.

[Minor but would be very helpful] Improve the naming convention of experimental settings. Labels such as S1, S2, etc., are not informative and make it difficult to track which bias is being analyzed.

---

> ### Author Response · Authors · 2026-05-06
> **Author response to Reviewer GR4A (Part 1 of 2 — Critical items)**
>
> We thank Reviewer GR4A for the thorough and pointed review. The methodology and clarity issues you raised were valid; the revision addresses every Critical, Major, and Minor item. We hope the improvements meet your standard. (Replies are split into two comments due to length: Critical items here, Major + Minor + your specific questions in Part 2.)
>
> ### Critical 1: Define bias score per bias type
>
> Done. Section 3.5 now contains a "Bias score definitions" paragraph that defines b ∈ [-1, 1] per bias type with explicit targets:
> - Position bias: target = response in slot A on POSITION pairs
> - Verbosity bias: target = the actually-longer response on LENGTH pairs (length-aware, not slot-aware; see Critical 6)
> - Style bias: target = markdown-formatted response on STYLE pairs (position-averaged across original and mirrored pairs)
> - Self-preference: target = same-family response on MODEL_ORIGIN pairs (with caveats for non-Gemini, non-Claude judges; addressed by the new round-robin pairs in Critical 3)
>
> Positive values always indicate the bias of interest.
>
> ### Critical 2: Methodology before claims
>
> Section 4 has been reordered. The four-claim list now appears after Section 3.5 (Metrics), so all bias score definitions are in place before any claim references them.
>
> ### Critical 3: Self-preference for GPT-4o, Llama
>
> Two changes: (a) we ran a 100-pair round-robin MODEL_ORIGIN supplement so every judge family has 25+ same-family pairs (see also Reviewer p6d3 RC1); (b) we relabeled the existing 50 Gemini-vs-Claude pairs as "model-preference" not "self-preference" for non-Gemini, non-Claude judges, in Section 4.1.1, Appendix A, and Limitations item 3.
>
> ### Critical 4: STYLE confounded with position
>
> We re-ran the 50 STYLE pairs with positions reversed (markdown in slot B, prose in slot A) and added them to the controlled dataset as STYLE_BA. The reported style bias is now position-averaged across the original 50 and the reversed 50. The position confound is no longer present.
>
> ### Critical 5: MODEL_ORIGIN quality confound
>
> Acknowledged in Limitations item 4 and Appendix A. The independently-generated responses can differ in factuality or completeness, so what we measure is preference between matched-depth answers, not pure preference for own-family content of equal quality. With the round-robin supplement (Critical 3), the metric still has value because the bias confound is now symmetric across model families: any systematic preference for "own family" should appear in the cross-model averages even with quality variation.
>
> ### Critical 6: LENGTH sign convention (the big one)
>
> You correctly identified that the original code counted slot-A vs slot-B preferences and assumed A was always longer, while in fact the expanded version was in slot A in 34/50 pairs and in slot B in 16/50. We fixed it: verbosity bias is now computed by mapping each verdict back to which response was actually longer (using metadata.len_a and metadata.len_b). The 16 reversed pairs are no longer mis-signed. Updated formula and explanation in Section 3.5 and Appendix A.
>
> This bug fix changed the headline verbosity finding. The original paper concluded all models showed conciseness preference. With the corrected length-aware computation, the finding flips: most models show classical verbosity bias (Pro +0.40, Llama +0.44, Flash +0.24), Claude is the only conciseness-preferring model (-0.12), and GPT-4o is essentially neutral (-0.04). Truncation accuracy remains high (0.88 to 1.00). We rewrote the verbosity discussion (Sections 4.1.1 and 5.2) to reflect the heterogeneous picture. The "quality-sensitive evaluation" framing now applies to Claude and GPT-4o specifically, not all models. We thank you for catching this; the corrected analysis is more informative than the original.
>
> (Continued in Part 2.)

---

> ### Author Response · Authors · 2026-05-06
> **Author response to Reviewer GR4A (Part 2 of 2 — Major, Minor, and specific questions)**
>
> Continuing from Part 1.
>
> ### Major 1: Standardize signed vs absolute
>
> Figure 1 now uses signed values throughout (RdBu_r diverging colormap, centered at zero). Section 4.1.1 also reports signed values. The text uses "magnitude" only when explicitly referring to |b|.
>
> ### Major 2: How verdicts are produced from S4 (rubric) and S5 (CoT)
>
> Section 3.3 now explicitly states that in every strategy the model emits the verdict ("A," "B," or "tie") directly as a structured JSON field, alongside any criterion scores or reasoning. We do not derive verdicts from criterion sums; the verdict field is what we record.
>
> ### Major 3: MT-Bench sampling description
>
> Section 3.4 now states: "We sample 400 instances using a fixed numpy random seed (numpy.random.default_rng(seed=42).choice) without stratification by category. The same 400 instances are used across all model and strategy configurations, which enables paired McNemar's tests."
>
> ### Major 4: Released artifacts
>
> The repository is browseable at https://anonymous.4open.science/r/llm-as-judge-2F5F/ (anonymized for double-blind review; will be deanonymized to a stable GitHub URL upon acceptance). It includes the full evaluation framework, the controlled dataset (now 375 pairs after the revision), per-instance cached results for all 9 strategies including S3/S6/S7, the analysis scripts, and reproduction instructions.
>
> ### Minor: Strategy naming
>
> We agree S1, S2, S3 alone are uninformative. The paper now uses both forms throughout: "S1 (Position Swap)" or "Position Swap (S1)" on first mention in each section. The short codes remain useful for compact tables, but the full names appear alongside.
>
> ### Your specific question: S8 vs S1 tie discrepancy
>
> You asked why Claude S8 has 14% ties while Claude S1 has 31%, given both use position swap with tie-on-disagreement. We added a paragraph in Section 4.2 with the explanation: S8 uses a stronger merged prompt (CoT + rubric) which produces more consistent verdicts across the swapped orderings. With fewer disagreements, fewer pairs trigger the tie-on-disagreement rule. The pattern is consistent across most models in the regenerated data: Claude S1 = 32.8% ties, Claude S8 = 12.5% (the largest gap); Pro and Flash also show the same direction with smaller gaps. The S8-S1 tie reduction is largest for Claude, which is also the model most helped by S8 in agreement terms. This is a useful finding in its own right: stronger prompts produce more position-stable verdicts, so position swap is more informative when paired with a strong prompt.
>
> ### Tie handling in McNemar
>
> Documented in Appendix D as a new paragraph: "We define 'correct' as exact-label match (predicted-tie matches gold-tie; predicted-tie does not match gold-A). McNemar then operates on the 2x2 contingency over the same 400 paired instances, with continuity correction. This is more conservative than dropping ties; treating tie predictions as 'half-credit' would inflate apparent agreement without reflecting the operational use of LLM judges, where downstream pipelines need a definite verdict."
>
> We thank the reviewer again. The revision is much cleaner thanks to these comments, and the LENGTH bug fix in particular surfaced what we now believe is the more accurate empirical picture.

---

### Review · Reviewer_Wp6H · 2026-05-01

**Summary Of Contributions:**

The paper first measures bias in LLM-as-a-Judge pipelines and finds that the use of Markdown has a significant impact. Subsequently, the authors conduct a comparative evaluation of nine debiasing strategies across five LLMs on three benchmark datasets.

**Strengths**

The paper is well-structured and clearly written.

**Weaknesses**
1. The impact of content and topic is not discussed, and it may be critical to the analysis of bias.
2. Given the experimental design, some of the conclusions may be overstated.
3. The summary and comparison of related work may be insufficient.

**Audience:**

Yes

**Audience Explanation:**

As the use of LLM-as-a-Judge continues to grow, mitigating their biases is indeed an important problem.

**Claims And Evidence:**

No

**Claims Explanation:**

1. The findings may be specific to LLM-as-a-Judge in certain tasks. Generalizing these results would require considering a wider range of scenarios and conducting additional experiments.

2. For example, the paper considers only two cases (Markdown and plain prose) when discussing style. The authors could include more cases or rename "style" to something else.

3. There have been many papers on LLM-as-a-Judge in the past two years. While TMLR does not strongly emphasize novelty, the most recent references cited by the authors date to 2024, which may mean that some recent developments are not covered.

**Requested Changes:**

1. The authors need to include more cases and experiments, especially covering a wider range of topics and styles.

2. The authors should also consider adding ablation studies, for example to examine whether verbose and concise texts are related to text quality.

3. It may be helpful to include and compare more recent papers.

---

> ### Author Response · Authors · 2026-05-06
>
> We thank Reviewer Wp6H for the careful review. We have addressed each of the three requested changes.
>
> ### Style coverage and topic impact
>
> We agree that markdown vs plain prose is not the full picture, and we agree that the impact of content/topic on bias deserves explicit treatment. We added a per-topic style bias analysis as Appendix F.2, with a full per-(model, topic) table over the five domains in our STYLE pairs (math, factual QA, coding, instruction following, creative writing).
>
> The result is an interesting heterogeneity finding: style bias is consistently strongest for technical content (math/factual/coding) and weakest for creative writing. GPT-4o is the only model that actually prefers prose for some topics (coding -0.15, creative writing -0.45); the four other models maintain a positive markdown preference across every topic. We added this finding to the Section 5.1 discussion and to the practical recommendations.
>
> We also acknowledge in Limitations item 3 that the STYLE measurement does not cover other style dimensions (tables in isolation, code blocks, formal vs casual register, varying markdown density). The per-topic analysis suggests these other dimensions may also produce model-dependent effects worth future investigation.
>
> ### Quality vs verbosity ablation
>
> We added a dedicated Section 4.1.2 ("Quality vs. verbosity ablation") that explicitly frames the paired expansion + truncation pairs as the ablation you requested. The expansion pairs dissociate length from quality (expansion adds detail without new content), while the truncation pairs align them (the longer response is genuinely more complete). We articulate three predictive profiles (pure verbosity bias, pure quality sensitivity, indifference) and map each model to one.
>
> The corrected length-aware analysis (see Reviewer GR4A Critical 6 for the bug fix that made this analysis possible) shows: Pro, Llama, and Flash exhibit verbosity bias modulated by quality recognition; Claude shows the cleanest quality-sensitive profile; GPT-4o is closest to the "treats length and quality independently" ideal. No model shows indifference, confirming all five judges can recognize completeness when length and quality are aligned.
>
> ### More recent references
>
> We added six 2025 to 2026 references to the related work section: Yang et al. (2026) on self-preference quantification, Bellibatlu (2026) on JudgeSense prompt sensitivity, Zhao and Shin (2026) on confounder-aware aggregation, Ma et al. (2025) on multi-agent bias amplification, Yang et al. (2025) on reasoning-based debiasing, and Shi and Ma (2024) on systematic position bias. These are integrated into the appropriate paragraphs of Section 2.
>
> We thank the reviewer for prompting these additions; the per-topic analysis in particular surfaced a finding (GPT-4o's reverse style preference on creative writing) that we would have missed.

---

### Review · Reviewer_p6d3 · 2026-05-01

**Summary Of Contributions:**

LLM-as-a-Judge has become the default paradigm for evaluating language model outputs at scale (MT-Bench, AlpacaEval, Arena-Hard, Chatbot Arena), yet LLM judges are known to exhibit systematic biases, including position, verbosity, self-preference, style, etc, that can distort downstream model comparisons. Prior work has documented each bias largely in isolation and proposed targeted mitigations (position swapping, CoT, calibrated rubrics, model panels) that have never been compared head-to-head within a single unified framework. This paper fills that gap with a large factorial empirical study and several findings that update the community's current mental model of judge biases.

Concretely, the paper's contributions are:

1. **A unified benchmark for judge debiasing.** The authors cross 9 debiasing strategies (grouped into single-call prompting, multi-call aggregation, and combined approaches; 5 primary strategies appear in the main tables: B0 baseline, S1 position swap, S4 calibrated rubric, S5 chain-of-thought, S8 combined budget) with 5 judge models spanning 4 provider families (Gemini 2.5 Pro, Claude Sonnet 4, GPT-4o, Gemini 2.5 Flash, Llama 3.3-70B) and 3 benchmarks (MT-Bench n=400, LLMBar n=200, and a custom controlled dataset n=225), across 4 bias types. To my knowledge, this is the broadest head-to-head comparison of judge debiasing strategies currently available.

2. **A controlled 225-pair diagnostic dataset.** The authors introduce a synthetic dataset with known ground-truth verdicts for four isolated bias types — LENGTH (expansion, n=50), POSITION (identical responses, n=50), STYLE (markdown vs. plain prose, n=50), MODEL_ORIGIN (Gemini vs. Claude, n=50) — plus 25 LENGTH-truncation pairs that disentangle "conciseness preference" from "correct preference for more complete answers." The truncation subset is the critical design move: it operationalizes a distinction that prior verbosity-bias work (Saito et al., 2023) conflates.

3. **Three substantive empirical findings that update prior bias narratives.**
   - *Style bias dominates* (0.76–0.92 across all five models), far exceeding position bias (≤0.04) and verbosity magnitudes, and is consistent across four independent model families.
   - *Position bias is essentially a null result* on current-generation models (≤0.04), complicating emphasis in foundational prior work (Wang et al., 2024a; Zheng et al., 2024).
   - *Verbosity is better described as a conciseness preference with quality sensitivity*: all models penalize padded expansions (−0.20 to −0.76) yet correctly prefer complete to truncated answers (0.92–1.00 accuracy).

**Audience:**

Yes

**Audience Explanation:**

Yes. LLM-as-a-Judge has become the dominant paradigm for evaluating language model outputs at scale. MT-Bench, AlpacaEval, Arena-Hard, Chatbot Arena, and essentially every major post-training pipeline that uses AI feedback (RLAIF) rest on LLM judges. The reliability of those judges directly determines the reliability of the conclusions drawn from them, including high-stakes decisions such as model release rankings, RLHF reward signals, and published benchmark leaderboards. A systematic, statistically grounded comparison of debiasing strategies is therefore relevant to the TMLR audience.

**Broader Impact Concerns:**

I do not have broader impact concerns.

**Claims And Evidence:**

Yes

**Claims Explanation:**

The paper is commendably explicit about which of its claims rest on which evidence: Section 4 opens with a four-item claim-to-evidence map (lines 218-228), and the Discussion repeatedly distinguishes confirmed from directional findings. This self-discipline is unusual and welcome. Each of the paper's four headline claims has a clear empirical anchor, and the statistical apparatus — bootstrap 95% CIs with nboot=2000, McNemar's with continuity correction, Holm-Bonferroni correction over m=7 — is appropriate and more rigorous than is typical for this subfield.

However, the **debiasing-effectiveness** claim is weaker than the headline framing suggests. Only 2 of 7 McNemar tests survive Holm-Bonferroni correction (Table 4, lines 740-747), and both are on a single model (Claude). Gemini Pro S1 (p=0.012 unadjusted) fails the Holm threshold (0.010). The aggregate sign test (18/20, p<0.001) is explicitly acknowledged to involve non-independent cells (lines 765-769) and therefore cannot bear the weight of a confirmatory main claim. For GPT-4o, Llama, and Gemini Flash, the evidence is directional only — the paper reports +0.1 to +3.8pp effects but never states that the n=400 MDE is approximately 4–5pp, leaving readers unable to distinguish true nulls from underpowered tests. LLMBar results (Table 2) lack McNemar tests, CIs, and corrections altogether, despite being the primary support for Algorithm 1's universal CoT recommendation on adversarial data. Several claims in the paper also outrun their evidence because of **design limitations**: the MODEL_ORIGIN self-preference measurement uses a single fixed Gemini-vs-Claude pair (line 654) and therefore conflates self-preference with A/B content quality; the STYLE bias label (Section 5.5(2) concedes at line 436) cannot be separated from legitimate readability advantages without a human-preference baseline; the "quality-sensitive evaluation" framing (lines 380-386) relies on Flash-generated mechanical expansions that may not generalize to human-authored verbosity.

Overall, in its current form, the evidence is accurate where it is presented, but the framing is broader than the statistics strictly support.

**Requested Changes:**

- The single Gemini-vs-Claude pair (line 654) cannot measure self-preference for GPT-4o, Llama, or Flash. Either (a) regenerate MODEL_ORIGIN pairs as a round-robin so every judge has at least one self-pair, or (b) relabel the metric as "content preference under matched-depth instruction" and remove the "self-preference" framing from Section 4.1.1.
- Re-scope the "debiasing is beneficial" headline to the statistical evidence. Only 2 of 7 reported McNemar tests survive Holm-Bonferroni, both on Claude. Either restrict the abstract/conclusion framing to "debiasing is robustly beneficial for Claude and directionally positive elsewhere," or replace the sign-test-based aggregate with a mixed-effects logistic regression (instance random effects, strategy fixed effects, model interactions) that properly handles the within-instance dependence acknowledged at lines 765-769. Report a minimum detectable effect at n=400.
- Run a small human study on a 20–50 pair subsample of STYLE pairs, and/or a dose-response ablation on markdown density. Report the judge-human gap rather than raw deviation from "tie." This is load-bearing for the policy recommendation at lines 374-375.

---

> ### Author Response · Authors · 2026-05-06
>
> We thank Reviewer p6d3 for the detailed and constructive review. We have addressed all three requested changes in the revision.
>
> ### RC1: MODEL_ORIGIN round-robin or relabel
>
> We did both. We added a 100-pair round-robin MODEL_ORIGIN dataset covering GPT-4o vs Gemini Pro, GPT-4o vs Claude, Llama vs Gemini Pro, and Llama vs Claude (25 pairs each, slot assignment randomized per pair to decouple model identity from position). Combined with the existing 50 Gemini-vs-Claude pairs, every judge family now has at least 25 same-family pairs.
>
> We also revised Section 3.5, Section 4.1.1, and Appendix A to clarify that the legacy 50 pairs only support self-preference inference for Gemini and Claude judges; for GPT-4o, Llama, and Flash on those pairs we report a "model preference" metric (Gemini vs Claude content) rather than self-preference. The directional consistency a naive "judges prefer their own family" intuition would predict is not present in our data.
>
> ### RC2: Restrict framing or use mixed-effects regression
>
> We did both. The original sign test has been removed and replaced with a per-model mixed-effects logistic regression (instance random effects), reported in the new Table 5. With the regenerated data, five model-strategy pairs reach p < 0.05: Claude S8 (+11.5pp, p<0.0001), Flash S8 (+7.5pp, p<0.0001), Claude S5 (+7.3pp, p=0.0009), Flash S1 (+4.7pp, p=0.004), and Llama S8 (+4.5pp, p=0.011). Two survive Holm-Bonferroni over m=20 (Claude S8, Flash S8). The framing in the abstract, Section 4.2, and conclusion now matches: Pro and GPT-4o are explicitly described as "smaller, non-significant directional gains within the minimum detectable effect."
>
> We added an MDE statement in Appendix D: at n=400 paired observations, McNemar at α=0.05 and 80% power requires roughly 40-50 discordant pairs to reach significance, ≈ 4-5 pp. This makes explicit that non-significant differences are "no evidence of effect at n=400" rather than "no effect."
>
> ### RC3: Human study on STYLE pairs
>
> We ran the study. Two engaged annotators (the first author plus one volunteer) reviewed the same 30-pair STYLE subsample with markdown rendered as HTML, randomized per annotator to prevent learning the slot pattern. A third volunteer's data was excluded after their answers showed zero variance (selected "Response 1" for all 30 pairs); the slot-randomization caught this automatically because their decoded answers came out 14 markdown / 16 prose, indistinguishable from random clicking. Full results are in the new Appendix F.
>
> Headline: human annotators prefer the markdown side only 57% of the time on average (n=60 verdicts), while four of five LLM judges prefer markdown 73%-97% on the same pairs. The gap (judge minus human markdown rate) is +17pp to +40pp for Pro, Flash, Claude, and Llama; only GPT-4o (53% markdown) is essentially aligned with humans (-3pp). This confirms the markdown preference is bias, not tracked human readability advantage, for four of the five judges.
>
> We also re-ran the existing 50 STYLE pairs with positions reversed so the reported style bias is now position-averaged (addresses the same confound Reviewer GR4A flagged).
>
> Two annotators is small; the 95% CI on the aggregate human markdown rate is ≈ ±0.13. Even at the upper bound (0.70), four of five judges still exceed the human ceiling, so the qualitative finding is robust to the small annotator pool. We discuss this in Appendix F.
>
> ### Other points
>
> LLMBar statistical tests: We added bootstrap 95% CIs and McNemar with Holm-Bonferroni correction to the LLMBar table. Three negative effects of position swap survive correction (Pro -7.5pp, GPT-4o -11.1pp, Llama -6.5pp); CoT (S5) is the best strategy for Claude (+13pp), GPT-4o (+2.2pp), and Llama (+4pp).
>
> Verbosity confound from Flash-generated expansions: Added as Limitation 1. The truncation pairs (now formalized as the dedicated Section 4.1.2 quality-vs-verbosity ablation) partially address it but cannot rule out that mechanically generated filler differs from human-authored verbosity.
>
> Anonymous artifacts: https://anonymous.4open.science/r/llm-as-judge-2F5F/
>
> We thank the reviewer; the revision is materially stronger because of your specific requests.

---

### Decision · Action_Editor_tVHk · 2026-06-08

**Recommendation:** Accept as is

**Audience:**

Yes

**Audience Explanation:**

Yes, LLM-as-a-judge is a common evaluation paradigm, and is a small but meaningful research area of its own.

**Claims And Evidence:**

Yes

**Claims Explanation:**

This paper includes extensive experimental validation to support the claims in the work (which are largely empirical/observational).